# Optimization of Growth Conditions of *Desulfovibrio desulfuricans* Strain REO-01 and Evaluation of Its Cd(II) Bioremediation Potential for Detoxification of Rare Earth Tailings

**DOI:** 10.3390/microorganisms13071511

**Published:** 2025-06-28

**Authors:** Ping Zhang, Chaoyang Wei, Fen Yang

**Affiliations:** 1School of Environmental Science And Engineering, Tongji University, Siping Rd. 1239, Shanghai 200092, China; zhangping0708@tongji.edu.cn; 2Key Laboratory of Land Surface Pattern and Simulation, Institute of Geographic Sciences and Natural Resources Research, Chinese Academy of Sciences, Beijing 100101, China; weicy@igsnrr.ac.cn

**Keywords:** ion-adsorption rare earth tailings, sulfate-reducing bacteria, metabolic activity of SRB, cadmium immobilization, carbon source optimization

## Abstract

To promote environmentally sustainable remediation and resource recovery from ion-adsorption rare earth tailings (IRET), this study comprehensively investigated the previously isolated strain REO-01 by examining its sulfate-reducing performance, Cd(II) immobilization potential, and physiological and biochemical responses under varying environmental conditions. Strain REO-01 was identified as a Gram-negative facultative anaerobe with strong sulfate-reducing activity and effective Cd(II) immobilization capacity. During a 96 h incubation period, the strain entered the exponential growth phase within 36 h, after which the OD_600_ values plateaued. Concurrently, the culture pH increased from 6.83 to 7.5, and the oxidation-reduction potential (ORP) declined to approximately −300 mV. Cd(II) concentrations decreased from 0.2 mM to 3.33 μM, corresponding to a removal efficiency exceeding 95%, while sulfate concentrations declined from 1500 mg/L to 640 mg/L, with a maximum reduction efficiency of 66.16%. The strain showed optimal growth at 25–40 °C and near-neutral pH (6–7), whereas elevated Cd(II) concentrations (≥0.2 mM) significantly inhibited cell growth. A sulfate concentration of 1500 mg/L was found to be optimal for cellular activity. Among the tested carbon sources, sodium lactate at 4.67 g/L yielded the most favorable results, reducing ORP to −325 mV, increasing pH to 7.6, and lowering Cd(II) and sulfate concentrations to 3.33 μM and 510 mg/L, respectively. These findings highlight the strong potential of strain REO-01 for simultaneous sulfate reduction and Cd(II) remediation, supporting its application in the in situ bioremediation and resource utilization of rare earth tailings.

## 1. Introduction

To facilitate leaching, the extraction of ion-adsorption rare earth ores relies heavily on chemical reagents, such as ammonium sulfate, to facilitate leaching processes [1]. This practice results in the accumulation of high sulfate concentrations in tailing areas, which in turn promotes the mobilization of heavy metals (e.g., Cd^2^^+^) from ore matrices [2]. These contaminants—particularly sulfate and heavy metals—can lead to soil acidification and groundwater pollution, thereby threatening surrounding farmland and aquatic ecosystems via leaching and hydrological dispersion. Conventional physicochemical remediation approaches, such as chemical precipitation, neutralization, and adsorption, have been applied to mitigate these environmental issues [3,4]. However, their limited efficacy, high operational costs, and the risk of secondary pollution often restrict large-scale application. In contrast, bioremediation, particularly through the application of SRB, has emerged as a sustainable and environmentally friendly alternative [5]. SRB are prokaryotic microorganisms capable of reducing sulfur oxyanions (e.g., sulfate, thiosulfate, and sulfite) to sulfide via anaerobic respiration [6]. The biogenic sulfide produced in this process readily reacts with heavy metals (e.g., Cd^2^^+^), leading to the formation of insoluble metal sulfide precipitates. In addition to metal immobilization, SRB activity improves tailing site conditions by increasing the pH and decreasing ORP.

SRB reduce sulfur compounds using either inorganic (chemolithoautotrophic) or organic (chemo-organotrophic) electron donors, ultimately generating sulfide [7,8,9]. Their metabolic versatility allows the oxidation of a wide array of substrates, including alcohols, short-chain fatty acids, and hydrogen [10]. Notably, certain SRB strains can also use alternative electron acceptors—such as ferric iron (Fe^3^^+^) and nitrate (NO_3_^−^)—under sulfate-deficient conditions [11]. These versatile metabolic pathways allow SRB to act as key agents in global biogeochemical cycles involving sulfur, carbon, nitrogen, and various metals. Therefore, the isolation and characterization of environmentally tolerant SRB from IRET not only provides novel microbial resources for the bioremediation of tailing-associated pollutants, but also broaden SRB applications, offering valuable insights into pollution mitigation and biogeochemical cycling.

SRB constitute a phylogenetically and metabolically diverse group of microorganisms that are capable of reducing sulfate and are widely distributed in various anaerobic ecosystems [12]. Phylogenetic and metabolic pathway analyses have revealed substantial physiological and biochemical diversity among SRB, enabling their survival in extreme environments such as petroleum reservoirs, paddy soils [13], wetlands, deep-sea hydrothermal vents [14,15], acid mine drainage systems, and marine sediments [16]. In these ecosystems, SRB play essential ecological roles in sulfate reduction, heavy metal immobilization, and the regulation of redox conditions. Their metabolic flexibility and environmental relevance underscore their importance as functional microorganisms in pollution remediation strategies. In terrestrial and aquatic environments, the biogeochemical cycling of sulfate and heavy metals is governed by a range of processes—including redox reactions, microbial activity, precipitation and dissolution, complexation, adsorption and desorption, and biomineralization—all of which are strongly influenced by environmental factors such as organic matter content, pH, and redox potential (Eh), with SRB functioning as a key driver [17].

In recent years, the development of bioremediation technologies based on SRB metabolism has offered efficient and cost-effective alternatives for removing toxic pollutants, making SRB a central focus of environmental remediation research. However, several challenges still limit the practical application of SRB, including strain-specific physiological traits, metabolic efficiency, electron donor preferences, and tolerance to environmental stressors—all of which significantly influence remediation outcomes. A deeper understanding of the physiological and biochemical traits of SRB in specific contaminated environments, along with their responses to key environmental factors, is essential for optimizing their use in pollution control. Previous studies have demonstrated the feasibility of using SRB for mine wastewater treatment and heavy-metal-contaminated soil remediation [18,19]. For example, Tuppurainen employed wheat straw as a carbon source to investigate SRB-mediated remediation of Zn^2+^ and sulfate ions in an anaerobic upflow reactor, with positive outcomes in both metal immobilization and sulfate reduction [20]. Nevertheless, SRB strains vary significantly in their metabolic profiles, growth requirements, and environmental resilience. Therefore, the targeted isolation and screening of efficient SRB strains from specific contaminated environments, combined with detailed investigations on their environmental adaptability and remediation mechanisms, are critical for enhancing their engineering applicability.

In this study, we systematically evaluated the physiological characteristics, environmental adaptability, and bioremediation potential of a previously isolated SRB (designated REO-01), originally obtained from IRET. First, its metabolic type and electron donor utilization capacity were analyzed to elucidate its energy metabolism under anaerobic conditions—providing foundational data for subsequent ecological and engineering studies. The strain’s growth curve was then assessed to characterize its lag, exponential, and stationary phases, offering guidance for optimizing cultivation conditions and inoculum application strategies. Additionally, the strain’s influence on environmental parameters such as pH, ORP, cadmium (Cd^2^^+^), and sulfate levels during growth was examined to assess its performance in complex polluted environments, particularly in heavy metal immobilization and sulfate reduction. Environmental factors such as temperature, pH, Cd^2^^+^ concentration, and sulfate concentration were further investigated to determine their effects on SRB metabolism and growth, with the aim of enhancing the strain’s survival and remediation efficiency in contaminated environments. Furthermore, carbon sources, as key electron donors, were evaluated for their impact on SRB performance. Although various substrates—such as lactate, ethanol, and glycerol—have been used in SRB applications, the optimal concentrations and specific metabolic pathways involved remain insufficiently understood. This study, therefore, explores the effects of different carbon sources on strain REO-01, with the aim of identifying the most effective electron donors and their optimal concentrations to enhance remediation efficiency and reducing operational costs. Overall, this research offers comprehensive insights into the mechanisms underlying the environmental adaptability and remediation potential of strain REO-01. These findings contribute valuable microbial resources and scientific evidence for the development of efficient, eco-friendly bioremediation technologies. Moreover, they expand the potential applications of SRB in pollution control, heavy metal stabilization, and biogeochemical cycling—providing meaningful support for future studies in environmental microbiology, biohydrometallurgy, and related disciplines.

## 2. Materials and Methods

### 2.1. Strain Source and Cultivation

To obtain a suitable SRB strain, five successive subcultures were performed in a modified Postgate C enrichment medium (Table 1), followed by five rounds of colony purification on a solid agar medium. A representative dissimilatory SRB was isolated from IRET and designated strain REO-01. Strain REO-01 exhibited a curved rod-shaped morphology, measuring approximately 1.5–4.0 μm in length and 0.5–1.0 μm in width. On solid medium supplemented with ferrous ions, the colonies initially appeared grayish-white but gradually turned black after three days of incubation due to the formation of ferrous sulfide (FeS), a typical metabolic product of SRB. The 16S rRNA gene of strain REO-01 was amplified via polymerase chain reaction (PCR), and the resulting sequence was subjected to BLAST analysis using BLASTn (NCBI, version 2.13.0). The results indicated that strain REO-01 shared over 97% sequence similarity with multiple *Desulfovibrio* species, with the highest similarity (99%) to *Desulfovibrio* strain AF098671.1. According to *Bergey’s Manual of Determinative Bacteriology* [21], members of the genus *Desulfovibrio* are organotrophic, Gram-negative, non-spore-forming, motile bacteria that perform anaerobic respiration by reducing sulfur or other oxidized compounds. These bacteria are facultative anaerobes equipped with polar flagella and contain cytochrome c_3_, a key component of their electron transport system.

A phylogenetic tree was constructed using MEGA X software (version 10.2.6) based on the 16S rRNA gene sequence of REO-01 and those of closely related strains. Phylogenetic analysis confirmed that strain REO-01 belongs to the species *Desulfovibrio desulfuricans*, which is taxonomically classified under the domain *Bacteria*, phylum *Proteobacteria*, class *Deltaproteobacteria*, order *Desulfovibrionales*, family *Desulfovibrionaceae*, and genus *Desulfovibrio*. Strain REO-01 has been deposited in the China General Microbiological Culture Collection Center (CGMCC) under accession number CGMCC No. 24267. The corresponding 16S rRNA gene sequence has been submitted to the National Center for Biotechnology Information (NCBI) under the GenBank accession number OL454786.1. Details regarding the isolation, cultivation, and molecular identification methods of REO-01 are described in a previous study by our research group [22].

### 2.2. Biochemical Characterization of Strain REO-01

Microorganisms possess distinct enzymatic systems and carbohydrate metabolic pathways, resulting in varied metabolic products depending on the substrates utilized [23,24]. Therefore, biochemical profiling based on microbial metabolic products is commonly employed to assist in bacterial identification [25]. To characterize the biochemical traits of strain REO-01, the isolate was first cultured in enrichment medium and incubated anaerobically at 30 °C for 3 days, during which the medium turned completely black, indicating sulfate reduction and FeS formation. After incubation, 1 mL of the culture was collected using a micropipette and transferred into various test tubes for standard physiological, biochemical, and carbohydrate fermentation assays, following the methods described by Benavides [26]. These tests included the following: hydrogen sulfide production, indole test, catalase test, *N*-acetylglucosamine broth test, oxidase test, Gram staining, urease test, methyl red test, sodium pyruvate utilization, rhamnose test, tartrate utilization, acetate utilization, glucose fermentation, mannitol fermentation, arabinose fermentation, sucrose fermentation, malonate utilization, fructose fermentation, and nitrate reduction. All inoculated tubes were incubated in anaerobic culture bags (Mitsubishi, Tokyo, Japan) at 30 °C in the dark for 3 days. The procedures followed the *Manual of Common Bacterial System Identification* in detail. Additionally, the API 20A identification test (bioMérieux, Marcy-l′Étoile, France) was conducted according to the manufacturer’s instructions, and metabolic profiling using the Biolog GN2 MicroPlate system (Biolog, Inc., Hayward, CA, USA) was performed following Biolog Inc.’s guidelines.

### 2.3. Growth and Metabolic Dynamics of Strain REO-01

Under optimal conditions, bacterial growth via binary fission follows an exponential pattern, typically producing a “J-shaped” growth curve. In laboratory cultures, bacterial growth typically follows an “S-shaped” curve due to the presence of three distinct phases: the lag phase, the exponential (log) growth phase, and the stationary phase, eventually followed by a death phase [27]. This growth pattern results from an initial adaptation period after inoculation, followed by nutrient depletion and the accumulation of inhibitory metabolites. To assess the growth and metabolic activity of strain REO-01, a 50 mL portion of a 3-day-old culture was centrifuged at 5000× *g* for 5 min. The cell pellet was washed three times with sterile phosphate-buffered saline (PBS) and resuspended in PBS. The suspension was then adjusted to an OD_600_ of 0.3 to prepare a standardized bacterial inoculum.

This suspension was inoculated at a 0.1% (*v*/*v*) ratio into 45 mL of fresh liquid medium contained in sterile 50 mL centrifuge tubes. The medium contained 1500 mg/L sulfate, and the initial pH was adjusted to 6.8 using sterile 1 M NaOH or HCl after autoclaving. To investigate the influence of Cd(II) on bacterial activity, a sterile cadmium stock solution (1000 μg/mL, Macklin, Seattle, WA, USA) was added to achieve a final Cd(II) concentration of 0.2 mM. The cultures were incubated statically at 30 °C under anaerobic and dark conditions. Samples were collected at 12-h intervals from 0 to 96 h. At each time point, OD_600_ was measured using a UV–Vis spectrophotometer (Unico UV-2800A, Unico, Shanghai, China); pH and ORP were measured using a Leici PHSJ-4F pH meter (Leici, Shanghai, China) and an Asone RGS71 ORP meter (As-One Corporation, Tokyo, Japan), respectively. Cd(II) and sulfate concentrations: determined by inductively coupled plasma optical emission spectrometry (ICP-OES, Agilent 5900, Santa Clara, CA, USA). To avoid interference from black FeS precipitates, FeSO_4_·7H_2_O was omitted from the medium used for OD_600_ measurements. A sterile medium without bacterial inoculation was used as the blank control. All experiments were conducted in triplicate to ensure reproducibility.

### 2.4. Optimizing Environmental Conditions for Enhanced Growth and Remediation Efficiency of Strain REO-01

Temperature influences the substrate utilization efficiency of microorganisms by affecting heterotrophic respiration, thereby altering substrate availability and composition [28]. These temperature-driven changes affect microbial metabolic activity, community structure, and the formation of metabolic “hotspots” [29]. Most known SRB are mesophilic, exhibiting optimal growth and sulfate reduction efficiency within a temperature range of 20 °C to 40 °C. To evaluate the temperature dependence of strain REO-01, the bacterium was cultivated at 25 °C, 30 °C, 35 °C, and 40 °C under controlled conditions. A standardized bacterial suspension (OD_600_ = 0.3) was inoculated at a 0.1% (*v*/*v*) ratio into 50 mL of sterile medium containing 1500 mg/L sulfate and adjusted to an initial pH of 6.8. The cultures were incubated statically under anaerobic and dark conditions for three days. The OD_600_ values were measured daily using a UV–Vis spectrophotometer to monitor growth. All experimental conditions were performed in triplicate.

The inhibitory effect of pH on SRB is complex, involving physiological stress responses and changes in enzyme activity. To assess the pH tolerance of strain REO-01, experiments were conducted at 30 °C with a sulfate concentration of 1500 mg/L. After sterilization, the initial pH of the culture media was adjusted to 3.0, 4.0, 5.0, 6.0, and 7.0 using sterile 1 M HCl or NaOH. A 0.1% (*v*/*v*) inoculum of the bacterial suspension was added to sterile 50 mL centrifuge tubes and incubated at 30 °C under anaerobic and dark conditions. Samples were collected every three days over a 9-day period, and OD_600_ values were recorded. All experiments were performed in triplicate.

To investigate the effect of Cd(II) on bacterial growth, strain REO-01 was cultured at 30 °C under anaerobic and dark conditions in media containing 0, 0.05, 0.1, 0.2, and 0.3 mM Cd(II), with a constant sulfate concentration of 1500 mg/L and an initial pH of 6.8. Cd(II) concentrations were adjusted using a 1000 μg/mL cadmium standard solution prior to sterilization. The inoculation ratio was 0.1% (*v*/*v*). Samples were collected every three days, and OD_600_ values were measured with a UV–Vis spectrophotometer. Each condition was studied in triplicate.

Sulfate reduction is a defining metabolic feature of SRB and forms the basis of their application in the treatment of sulfate-rich wastewater. To evaluate the sulfate utilization capacity of strain REO-01, sulfate concentration gradients of 500, 1000, 1500, and 2000 mg/L were tested. The culture medium was adjusted to each concentration using a 1 M Na_2_SO_4_ stock solution before sterilization. Other conditions were maintained at constant conditions, as follows: incubation temperature of 30 °C, initial pH of 6.8, an anaerobic and dark environment (created using Mitsubishi Gas-Culture Anaerobic Bags and anaerobic culture pouches), and an inoculation ratio of 0.1% (*v*/*v*). Samples were taken every three days for OD_600_ measurements. All experiments were carried out in triplicate.

### 2.5. Comparative Effects of Sodium Lactate, Glycerol, and Ethanol on the Sulfate Reduction Performance of Strain REO-01

This experiment evaluated the influence of three carbon sources—sodium lactate (L group), glycerol (G group), and ethanol (E group)—on the performance of strain REO-01, specifically regarding sulfate reduction, Cd(II) immobilization, pH changes, and ORP in the culture system. The amounts of each carbon source were calculated based on their electron-donating capacities and the theoretical molar ratios of an electron donor to sulfate, corresponding to carbon-to-sulfate molar ratios of 1:1, 2:1, 3:1, and 4:1. These ratios corresponded to COD-to- [SO_4_^2−^] values of approximately 0.67, 1.32, 2.01, and 2.68, respectivley. The sulfate concentration was fixed at 1500 mg/L, and the specific amounts of each carbon source at the different carbon-to-sulfur ratios (CSR) were summarized in Table 2. The initial Cd(II) concentration and pH were set at 0.2 mM and 6.8, respectively. A carbon-free control group (CK) was included as a reference. A 0.1% (*v*/*v*) bacterial suspension of REO-01 was inoculated into 50 mL sterile culture tubes containing the prepared medium. The cultures were incubated statically at 30 °C under anaerobic and dark conditions. Samples were collected every 12 h to analyze pH, ORP, and the concentrations of sulfate and Cd(II). All experiments were performed in triplicate.

### 2.6. Statistical Analysis

All experimental data were obtained from at least three independent replicates and are expressed as mean ± standard deviation (SD). Statistical differences among the treatment groups were analyzed using one-way analysis of variance (ANOVA), followed by Tukey’s post hoc test for pairwise comparisons. A *p*-value < 0.05 was considered statistically significant. The key performance indicators evaluated included cadmium stabilization efficiency (%), sulfate removal rate (%), ORP (mV), and pH. For time-series data such as sulfate reduction and ORP changes, repeated measures ANOVA was applied when appropriate. All data analyses and visualizations were performed using OriginPro 2023 (OriginLab Corporation, Northampton, MA, USA) and IBM SPSS Statistics 26.0 (IBM Corp., Armonk, NY, USA). Figures were finalized using Adobe Illustrator 2023 (Adobe Inc., San Jose, CA, USA) to ensure publication-quality graphics.

## 3. Results and Discussion

### 3.1. Biochemical Characteristics and Bioremediation Capacity of REO-01

The REO-01 strain exhibited distinct physiological and biochemical characteristics (Table 3). It tested positive for hydrogen sulfide and indole production, catalase activity, and utilization of *N*-acetylglucosamine, but tested negative for oxidase and urease activity, the methyl red test, and Gram staining. In terms of carbon source utilization, REO-01 could metabolize sodium pyruvate, glycerol, tartrate, and acetate, but was unable to utilize glucose, mannitol, arabinose, sucrose, malonate, or fructose. Additionally, it demonstrated nitrate-reducing activity. These results are consistent with descriptions in *Bergey’s Manual of Determinative Bacteriology* (8th edition) and align with those of previous reports [30].

Figure 1a presents the 96-h growth curve of REO-01 in the sulfate-reducing system. The strain adapted well to the liquid culture, showing a short lag phase (0–12 h). A logarithmic growth phase occurred between 12 and 36 h, marked by rapid cell division and peak metabolic activity at 36 h. From 36 to 48 h, growth slowed, entering a stationary phase where cell numbers remained high, but metabolic activity declined. After 48 h, a gradual decrease in OD_600_ indicated the onset of the death phase, likely due to cell lysis and the accumulation of toxic metabolic byproducts. During sulfate reduction, SRB produce H_2_S, a gaseous compound that can diffuse across cell membranes and dissociate into S^2^^−^ in aqueous environments, exerting cytotoxic effects [31]. Notably, the culture was not purged with N_2_ to establish strict anaerobic conditions; nevertheless, REO-01 exhibited normal growth. This suggests that REO-01 is not an obligate anaerobe and likely possesses a high degree of oxygen tolerance. Such tolerance may be attributed to the presence of the *cyd* gene, which encodes a terminal reductase that enables bacterial survival under microaerobic conditions [32].

The pH variation of the system over 96 h is shown in Figure 1b. The pH decreased from 6.83 to 6.05 during the first 24 h, likely due to the generation of organic acids, such as acetic acid, during carbon source oxidation. This initial acidification may have imposed stress on the cells. As the reaction progressed, the pH gradually increased, reaching a neutral level (approximately pH 7.0) by 36 h and peaking at 7.5 by 60 h, after which it stabilized. The increase in pH is primarily attributed to the formation of gaseous H_2_S from H^+^ and S^2−^, which then escapes from the system, thereby reducing proton concentration. At a higher pH levels, H_2_S predominantly exists in the form of HS^−^ or S^2−^, which are less volatile, thereby buffering further pH increases and helping maintain near-neutral conditions. The ORP variation is shown in Figure 1c. During the first 24 h, the ORP remained around −100 mV, consistent with the lag phase. This redox potential (~−100 mV) represents the minimum threshold typically required to initiate sulfate reduction [33]. From 24 to 48 h, the ORP dropped sharply and stabilized around −300 mV, indicating a highly reducing environment favorable for sulfate reduction. This decline is attributed to the accumulation of reductive metabolites such as H_2_S.

Figure 1d shows the cadmium immobilization efficiency of REO-01. In the first 24 h, the Cd(II) concentration declined slowly, followed by a rapid drop between 24 and 36 h, and a more gradual reduction thereafter. By 96 h, the final Cd(II) concentration reached 3.33 μM, corresponding to a removal efficiency exceeding 95%. These results demonstrate the strong bioremediation capability of REO-01 toward Cd(II). A previous study by our group [22] confirmed the formation of FeS and CdS precipitates in the system through SEM-EDS and XRD analyses, suggesting that REO-01 facilitates the biomineralization of metal sulfides [34]. Furthermore, investigations into the extracellular polymeric substances (EPS) of REO-01 revealed that functional groups such as C=O, N–H, and C–N in cell surface proteins were involved in binding Cd(II) and Fe(II) [35,36]. Other key functional groups involved in Cd(II) binding included glycosidic linkages [37], hydroxyl [38], carboxyl [39], methyl [40], phosphate ester [39], and thiol groups [41]. Notably, at low Cd(II) concentrations, thiol groups exhibited the highest binding affinity for Cd(II) [22].

The sulfate reduction efficiency of REO-01 is shown in Figure 1e. During the initial 24 h, the reduction rate was low, consistent with the lag phase. From 24 to 60 h, the sulfate concentration dropped rapidly to 640 mg/L, correlating with the logarithmic growth phase and intense microbial activity. After 60 h, the sulfate reduction rate plateaued. By 96 h, sulfate removal efficiency reached approximately 66.16%. Previous studies have shown that sulfate reduction by SRB primarily occurs within the first 60 h, after which the process slows due to the depletion of electron donors and the accumulation of toxic sulfides [42,43].

### 3.2. Growth Response of Strain REO-01 to Environmental Factors

The growth of strain REO-01 over five days under four different temperature conditions in the sulfate reduction system is presented in Figure 2a. At 25, 30, 35, and 40 °C, the OD_600_ values from day 2 to day 5 ranged from 0.59 to 0.65, 0.36 to 0.48, 0.35 to 0.44, and 0.31 to 0.41, respectively. Previous studies have shown that SRB are capable of growth and metabolism across a wide temperature range; however, most are classified as mesophiles, with optimal anaerobic respiration occurring between 25 and 40 °C [44]. Within this experimental range, strain REO-01 appeared to maintain membrane integrity and transport capacity by modulating fatty acid composition and phospholipid profiles [45]. As a result, no statistically significant differences in OD_600_ were detected across the four temperature treatments (*p* > 0.05). In all systems, the OD_600_ values gradually declined over time, consistent with the growth pattern previously described for strain REO-01.

The effect of initial pH on the growth of strain REO-01 is shown in Figure 2b. By day 2 of sulfate reduction—corresponding to the exponential phase—the OD_600_ values under initial pH conditions of 7, 6, 5, 4, and 3 were 0.64, 0.56, 0.47, 0.31, and 0.29, respectively. Notably, growth patterns at pH 3 and 4 were nearly identical, with no statistically significant differences observed between them from days 3 to 5 (*p* > 0.05). In contrast, statistically significant differences (*p* < 0.05) in OD_600_ were observed among pH 7, 6, and 5 on day 2, but these differences diminished in later stages (*p* > 0.05). Neutrophilic SRB exhibit optimal sulfate-reducing activity at pH 7.0–7.5 [46,47]. At pH < 5, high proton concentrations generate substantial osmotic pressure across the cell membrane, forcing cells to expend energy to maintain intracellular pH through active proton export [48]. This additional energy burden reduces the resources available for cellular growth, thus inhibiting metabolic activity [49]. Furthermore, microbial viability typically declines sharply in environments with pH < 2 [50].

The influence of Cd(II) on the growth of strain REO-01 is illustrated in Figure 2c. On day 2, OD_600_ values under Cd(II) concentrations of 0, 0.05, 0.1, 0.2, and 0.3 mM were 0.64, 0.56, 0.34, 0.26, and 0.04, respectively. As the Cd(II) concentration increased, REO-01 activity was progressively inhibited, with nearly a complete loss of viability observed at 0.3 mM. Statistically significant differences (*p* < 0.05) in growth were observed among all treatments. Previous studies have shown that sulfide (S^2−^) generated during SRB metabolism reacts with Cd(II) to form CdS nanoparticles. Under light conditions, these nanoparticles can enhance nitrogen fixation in heterotrophic bacteria and promote biomass accumulation [51,52]. However, this study was conducted in the absence of light, precluding the confirmation of such effects. Nevertheless, the 0.05 mM Cd(II) treatment had a minimal inhibitory effect, suggesting that low Cd(II) concentrations may be tolerable. Under anaerobic and illuminated conditions, CdS nanoparticles can catalyze water photolysis, generating OH^−^ and increasing system pH [53]. However, the oxidative stress associated with CdS nanoparticle accumulation should not be underestimated [36]. Future investigations will explore the effects of Cd(II) on REO-01 activity under light exposure.

The response of REO-01 to varying sulfate concentrations is depicted in Figure 2d. On day 2, corresponding to the exponential phase, OD_600_ values for 500, 1000, 1500, and 2000 mg/L sulfate treatments were 0.46, 0.62, 0.65, and 0.58, respectively. These results suggest that 1500 mg/L sulfate supported the highest cellular activity, although no significant differences were found when compared to the 1000 and 2000 mg/L treatments (*p* > 0.05). The lowest growth was observed under 500 mg/L. Insufficient sulfate may lead to an imbalance between sulfate availability and electron donors, potentially resulting in feast-and-famine cycles [54], while excessively high sulfate levels could lead to over-oxidation of carbon sources and induce metabolic stress. The effects of the CSR on REO-01 physiology will be investigated in future studies.

### 3.3. Performance Evaluation of Lactate, Glycerol, and Ethanol in Sulfate-Reducing Bioremediation

SRB are capable of oxidizing a variety of electron donors, including ethanol [36] and glycerol [55], reflecting their diverse metabolic capabilities [5]. The efficiency of sulfate reduction depends significantly on carbon source [56], as different substrates can significantly influence sulfur transformation processes [57]. Furthermore, variations in the molecular weight and chemical structure among carbon sources may significantly affect their metabolic pathways in SRB [58]. The addition of easily biodegradable organic compounds has been shown to enhance the pollutant-degrading capacity of SRB. This section evaluates the effects of different carbon sources (sodium lactate, glycerol, and ethanol) on key performance indicators in sulfate reduction bioremediation systems, including pH, ORP, sulfate reduction efficiency, and Cd(II) immobilization.

Figure 3 illustrates the pH changes over 96 h in systems with different carbon sources. In the control (CK) group without added carbon sources, the pH remained close to the initial value of 6.8 throughout the experiment. In contrast, all three carbon sources led to increases in pH, although the extent of the change varied depending on the concentration. pH variation in sulfate-reducing systems is attributed to the equilibrium among reaction products such as H_2_S, HS^−^, S^2−^, CO_2_, HCO_3_^−^, and CO_3_^2−^ [59]. SRB can oxidize lactate, glycerol, and ethanol into acetate, which is subsequently be oxidized to HCO_3_^−^, consuming H^+^ and thereby increasing the pH of the system [60]. The redox reactions involved are represented as follows:(1)$$2{CH}_{3}CHOH{COO}^{-}+{{SO}_{4}}^{2-}\to 2{CH}_{3}{COO}^{-}+2{{HCO}_{3}}^{-}+{HS}^{-}+H^{+}$$(2)$$\begin{array}{c} {4CH}_{2}OHCHOH{CH}_{2}OH+3{{SO}_{4}}^{2-}\to \\ {4CH}_{3}COOH+4{H_{2}CO}_{3}+3{OH}^{-}+3{HS}^{-}+H_{2}O \end{array}$$(3)$$2{CH}_{3}{CH}_{2}OH+{{SO}_{4}}^{2-}\to 2{CH}_{3}{COO}^{-}+{HS}^{-}+H^{+}+2H_{2}O$$(4)$${CH}_{3}{COO}^{-}+{{SO}_{4}}^{2-}\to 2{{HCO}_{3}}^{-}+{HS}^{-}$$(5)$${{HCO}_{3}}^{-}+H^{+}\to {CO}_{2}+H_{2}O$$(6)$${H_{2}CO}_{3}+4H_{2}\to {CH}_{4}+{3H}_{2}O$$

Among the three carbon sources, sodium lactate (4.67 g/L) and ethanol (1.64 g/L), both with a CSR of 1.32, resulted in the highest final pH of 7.6, while glycerol (1.46 g/L, CSR 2.01) increased the pH to 7.3. In the lactate and glycerol systems, the pH initially decreased during the first 24 h and gradually increased thereafter. No significant differences in pH increase were observed among different concentrations (*p* > 0.05). In the ethanol system, the pH declined within the first 12 h and then increased, with the low-concentration treatments exhibiting significantly greater pH recovery than those with higher concentrations. The minimum pH values observed were as follows:: sodium lactate: 6.15–6.5; glycerol: 5.5–6.21; and ethanol: 6.5–6.7. The initial pH drop was attributed to acetate accumulation from incomplete substrate oxidation. Rapid acidification in glycerol systems may cause cellular damage and inhibit REO-01 growth [61], indicating that glycerol is less favorable than lactate or ethanol. Additionally, acidification may result from the reaction between Fe^2^^+^ and sulfide, releasing protons:(7)$${Fe}^{2+}+H_{2}S\to FeS(↓)+{2H}^{+}$$

The buffering capacities varied among the carbon sources. During the primary oxidation stage, lactate generates HCO_3_^−^ (Equation (1)), which helps maintain system neutrality [62]. In contrast, glycerol and ethanol produce HCO_3_^−^ only during secondary oxidation reactions (Equation (4)). Thus, lactate provides superior buffering, better supporting REO-01 metabolism and system stability. The increase in pH will be beneficial for the growth and metabolism of strain REO-01, thereby promoting sulfate reduction in the system. The limited production of HCO_3_^−^ when glycerol and ethanol are used as carbon sources may be insufficient to neutralize the H^+^ released during sulfate reduction [63], resulting in a pH decrease in the system.

Figure 4 shows the ORP trends across the different systems. In the control group, the ORP increased from an initial anaerobic level of approximately −100 mV to +200 mV, indicating a shift toward an oxidizing environment. In contrast, all carbon-supplemented systems maintained an ORP below −100 mV, except for the 1.95 g/L glycerol treatment, in which the ORP rose to −50 mV, potentially inhibiting SRB metabolism. Studies have shown that SRB can maintain normal metabolism at ORP levels below –100 mV; however, metabolism is inhibited when ORP exceeds this threshold [64]. Therefore, using glycerol as a carbon source is unfavorable for the growth and metabolism of SRB.

Among the three carbon sources, the sodium lactate system exhibited a generally decreasing ORP trend, with lower ORP values observed at moderate concentrations (Figure 4a). At 4.67 g/L, the ORP reached a minimum of −325 mV, providing favorable conditions for sustained sulfate reduction. However, concentrations higher than 4.67 g/L led to slower declines in ORP, likely due to substrate inhibition. Similarly, the rate of ORP decrease at a sodium lactate concentration of 2.34 g/L was slightly lower than that at 4.67 g/L, possibly due to the rapid depletion of carbon sources at lower concentrations. In the glycerol system (Figure 4b), the ORP initially declined and then gradually increased over time, displaying smaller overall fluctuations. The 1.46 g/L glycerol treatment achieved a minimum ORP of −230 mV at 72 h and maintained lower ORP levels than those observed at other glycerol concentrations. It is possible that moderate oxygen exposure in this system promoted the partial oxidation of sulfide, thereby reducing its toxicity [65], while, simultaneously, particle aggregation maintained anaerobic niches suitable for SRB activity [66]. In the ethanol system (Figure 4c), the ORP steadily decreased at lower concentrations (0.82 and 1.64 g/L), with the 1.64 g/L treatment reaching a minimum of −250 mV. Lower reduction potential is favorable for maximizing sulfate reduction efficiency [67]. In contrast, higher concentrations of ethanol (2.47 and 3.29 g/L) maintained ORP near −100 mV, which may not provide optimal redox conditions for SRB proliferation and activity. Taken together, these results suggest that the most effective ORP-lowering concentrations were 4.67 g/L for sodium lactate, 1.46 g/L for glycerol, and 1.64 g/L for ethanol.

Figure 5 presents the temporal changes in Cd(II) concentration across different systems supplemented with different carbon sources. In the control group, Cd(II) remained stable at approximately 0.2 mM, indicating negligible removal in the absence of external carbon input. In contrast, all carbon treatments promoted Cd(II) immobilization to varying extents, reflecting enhanced SRB activity. At their optimal concentrations—4.67 g/L (lactate), 1.46 g/L (glycerol), and 1.64 g/L (ethanol)—the systems achieved final Cd(II) concentrations of 3.33 μM, corresponding to CSR values of 1.32, 2.01, and 1.32, respectively. Among the treatments, the glycerol system showed a particularly rapid decline in Cd(II) concentrations across all tested dosages, suggesting an immediate and efficient bioremediation response. In contrast, the lactate and ethanol systems exhibited a more gradual decrease in Cd(II), particularly at higher concentrations, likely due to a delayed onset of sulfate reduction and subsequent sulfide production required for Cd precipitation. The final Cd(II) concentrations ranged from 3.33 to 9.99 μM in the sodium lactate and ethanol systems, and from 3.33 to 7.33 μM in the glycerol system. Notably, all three carbon sources achieved over 95% Cd(II) removal efficiencies, and no statistically significant differences were observed among the three treatments (*p* > 0.05). These findings confirm that lactate, glycerol, and ethanol are all effective carbon sources for promoting SRB-mediated Cd(II) stabilization under the tested conditions.

Figure 6 illustrates the sulfate concentration dynamics across different treatments over time. In the control group, sulfate remained constant at approximately 1500 mg/L, confirming the lack of sulfate-reducing activity in the absence of added carbon sources. All tested carbon sources significantly promoted sulfate reduction, though with varying degrees of efficiency and timing. Among the treatments, sodium lactate at 4.67 g/L yielded the most pronounced sulfate reduction, with a final sulfate concentration of 510 mg/L and a CSR of 1.32. Glycerol at 1.46 g/L and ethanol at 1.64 g/L resulted in final sulfate concentrations of 550 mg/L and 600 mg/L, respectively, with corresponding CSRs of 2.01 and 1.32. In the lactate system, sulfate reduction occurred rapidly at all concentrations, suggesting that lactate is readily utilized by REO-01 and effectively supports sulfate-reducing activity. Previous studies have also reported that lactate is considered the most suitable electron donor [68]. In contrast, both the glycerol and ethanol systems exhibited a lag phase, with significant sulfate reduction only being observed after 48 h of incubation, suggesting that these carbon sources may delay the onset of sulfate reduction [69]. Nevertheless, increasing the concentrations of sodium lactate and ethanol generally enhanced sulfate reduction efficiency. This trend is likely due to improved substrate availability, as an insufficient carbon source can impair both the sulfate reduction rate and the stability of its reduction products [70]. Furthermore, a limited electron donor supply restricts the physiological function of heterotrophic microorganisms, including SRB [71].

Interestingly, sulfate reduction in the glycerol system was highest at moderate concentrations, whereas higher concentrations were less effective. This may be attributed to the fact that glycerol metabolism can result in methane production (Equation (6)), making it a more suitable substrate for methanogenic archaea than for SRB. Consequently, excessive glycerol may stimulate methanogen proliferation, which competes with SRB for available electron donors and thereby suppresses sulfate reduction. In the lactate and ethanol systems, higher concentrations enhanced sulfate reduction, likely due to increased substrate availability, without triggering significant microbial competition. The sulfate reduction rate reflects the activity of the SRB in the reaction system [61]. These results indicate that lactate is the most effective carbon source for supporting sulfate reduction by strain REO-01.

The theoretical COD/SO_4_^2−^ ratio required for complete sulfate reduction is 0.67. However, actual systems often require higher ratios to achieve efficient performance [67]. Based on the observed effects of carbon sources on pH, ORP, sulfate reduction, and Cd(II) stabilization, the optimal CSR values determined in this study were 1.32 for sodium lactate, 2.01 for glycerol, and 1.32 for ethanol. These findings align with previous studies reporting that SRB perform optimally within a COD/SO_4_^2−^ range of 0.7 to 1.5, beyond which they are outcompeted by other microbial populations. In the present study, sodium lactate and ethanol followed this general pattern, whereas the glycerol system deviated, exhibiting a higher COD/SO_4_^2−^ ratio and resulting in lower sulfate reduction efficiency. As indicated in Equation (6), glycerol metabolism can lead to methane formation, which supports methanogen growth and hinders SRB competitiveness [56]. Moreover, strain REO-01 displayed relatively low adaptability to ethanol, requiring a longer acclimation period before efficient sulfate reduction was observed.

## 4. Conclusions

In this study, a previously isolated SRB, designated REO-01 and obtained from an IRET site, was comprehensively evaluated for its physiological characteristics and bioremediation potential. The strain exhibited typical physiological and biochemical characteristics of SRB, including strong sulfate-reducing capacity, detectable nitrate-reducing activity, and the ability to metabolize a broad spectrum of carbon sources. During 96 h of incubation, REO-01 demonstrated rapid growth and metabolic activity, effectively regulating environmental parameters such as pH and ORP to maintain favorable reducing conditions. Notably, REO-01 achieved over 95% Cd(II) immobilization and up to 66.16% sulfate removal, highlighting its strong potential for simultaneous remediation of heavy metal and sulfate contamination. Environmental adaptability tests further revealed that REO-01 maintained high activity and stability across a range of conditions, including temperatures of 25–40 °C, initial pH values of 5–7, and Cd(II) concentrations up to 0.1 mM. The strain exhibited optimal performance at a sulfate concentration of 1500 mg/L. Evaluation of carbon sources showed that sodium lactate (4.67 g/L), glycerol (1.46 g/L), and ethanol (1.64 g/L) all supported sulfate reduction by REO-01. Among them, sodium lactate was the most effective, promoting lower ORP, higher pH, enhanced Cd(II) stabilization, and more efficient sulfate reduction.

In summary, REO-01 is a neutrophilic SRB with notable environmental resilience and excellent bioremediation performance, particularly for the co-treatment of heavy metals and sulfate in rare earth tailing environments. These findings provide a valuable microbial candidate and theoretical foundation for the development of in situ bioremediation strategies. For future field applications, we propose enhancing REO-01′s survival and metabolic activity by immobilizing the strain on carbon-based carriers to form microbial granules. This approach may facilitate the establishment of a stable and efficient in situ bioremediation system, offering a practical and innovative solution for remediating complex contamination in rare earth tailings.

## Figures and Tables

**Figure 1 microorganisms-13-01511-f001:**
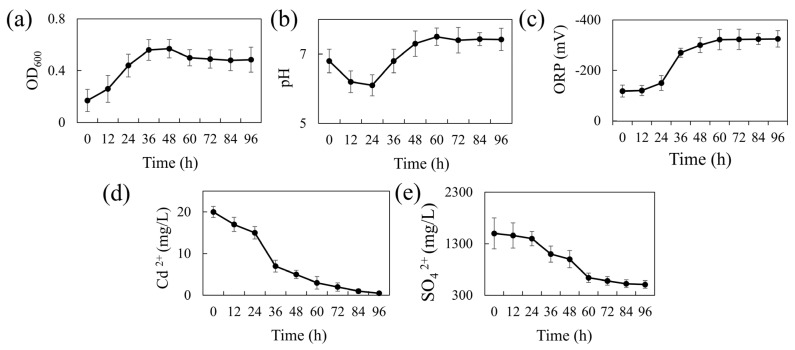
Temporal changes in REO-01 performance: (**a**) growth curve (OD_600_); (**b**) pH; (**c**) ORP; (**d**) Cd(II) concentration; (**e**) sulfate concentration.

**Figure 2 microorganisms-13-01511-f002:**
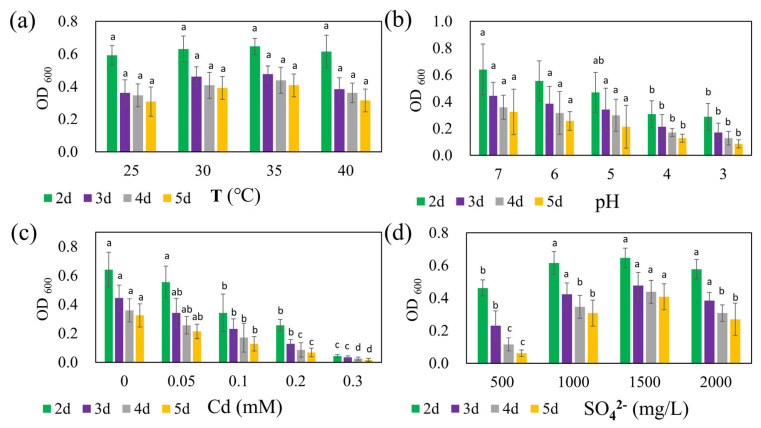
Effects of (**a**) temperature, (**b**) pH, (**c**) Cd(II) concentration, and (**d**) sulfate concentration on the growth of strain REO-01. Different lowercase letters (a, ab, b, c, d) above the bars indicate significant differences among treatments (*p* < 0.05, one-way ANOVA followed by Tukey’s test).

**Figure 3 microorganisms-13-01511-f003:**
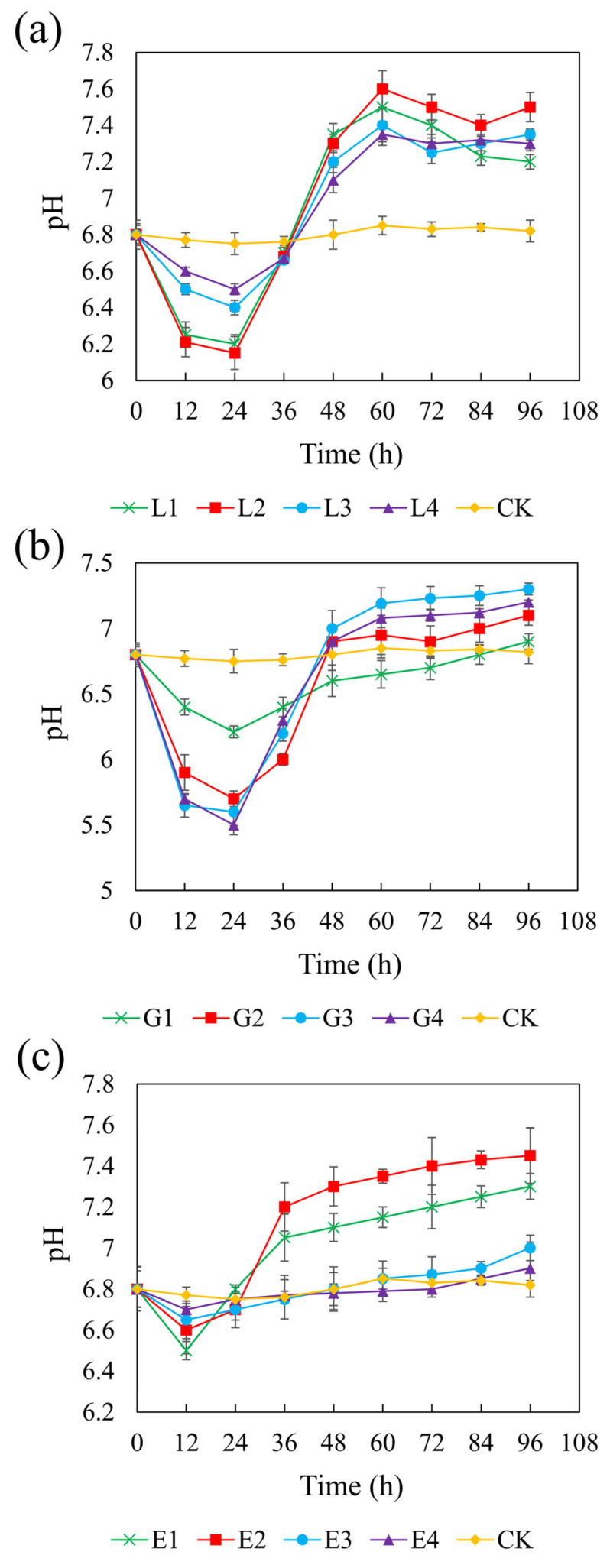
Effect of sodium lactate (**a**), glycerol (**b**), and ethanol (**c**) on pH in the REO-01 sulfate-reducing system.

**Figure 4 microorganisms-13-01511-f004:**
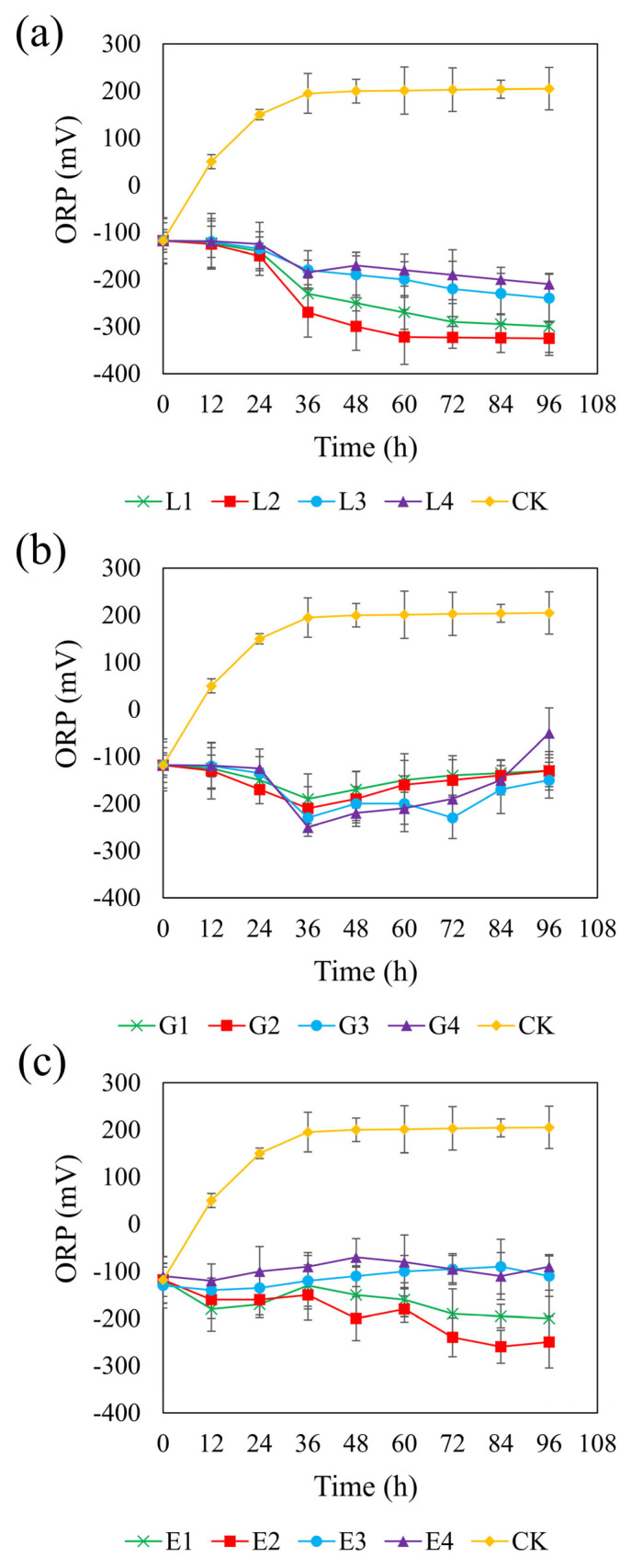
Effect of sodium lactate (**a**), glycerol (**b**), and ethanol (**c**) on ORP in the REO-01 sulfate-reducing system.

**Figure 5 microorganisms-13-01511-f005:**
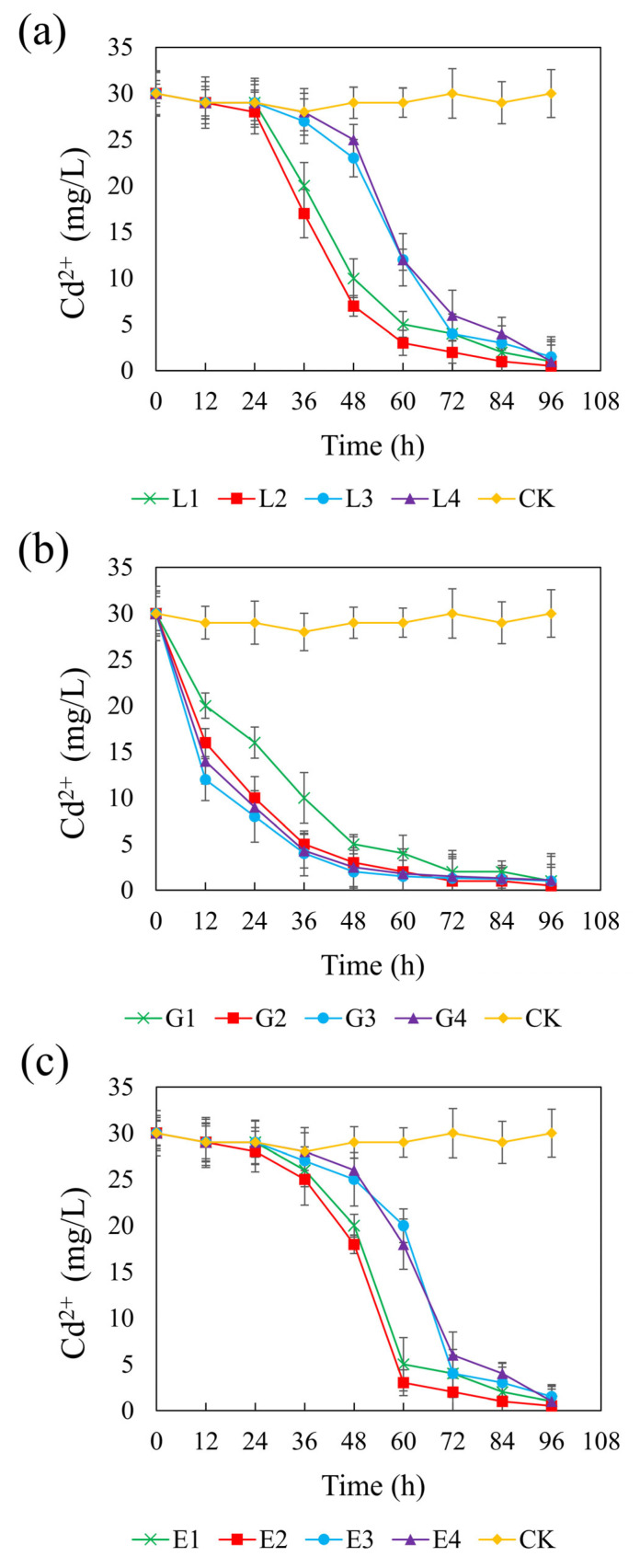
Effect of sodium lactate (**a**), glycerin (**b**), and ethanol (**c**) on cadmium stabilization efficiency in the REO-01 sulfate-reducing system.

**Figure 6 microorganisms-13-01511-f006:**
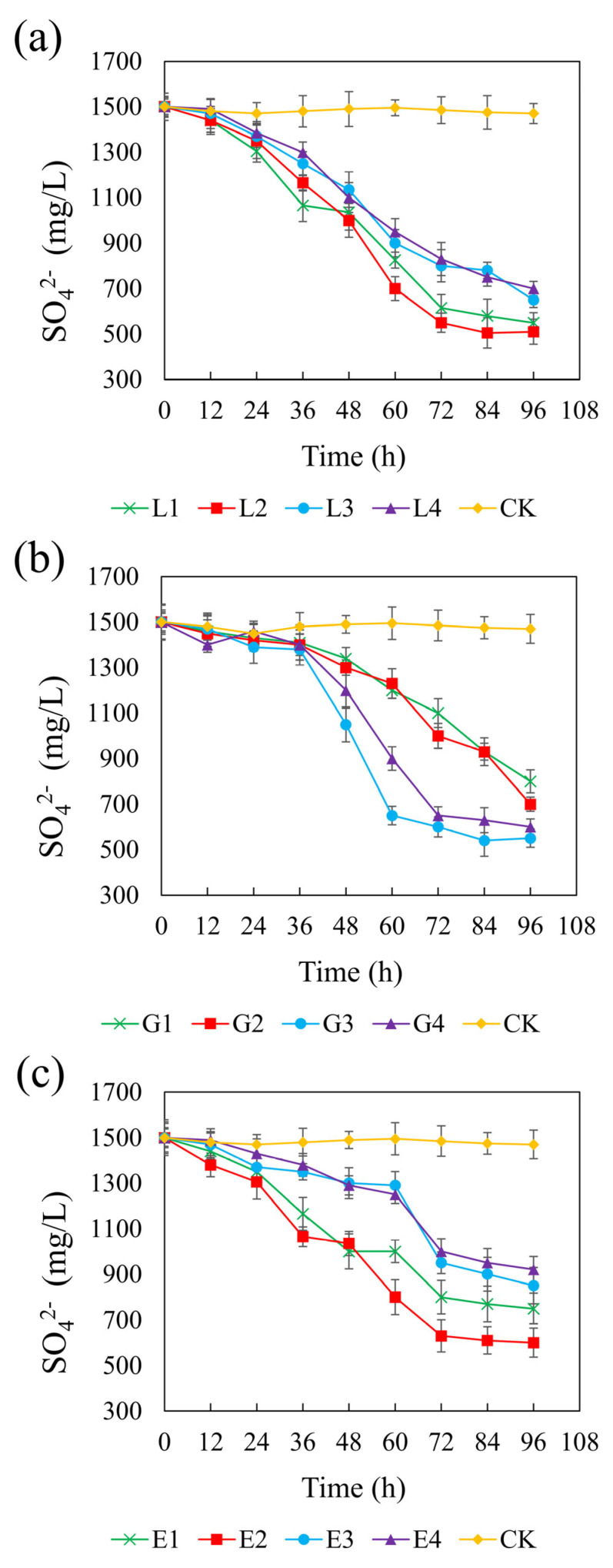
Effect of sodium lactate (**a**), glycerin (**b**), and ethanol (**c**) on the sulfate reduction rate in the REO-01 sulfate-reducing system.

**Table 1 microorganisms-13-01511-t001:** Composition of the enrichment medium used for SRB isolation.

Component	Concentration (g/L)	Component	Concentration (g/L)
KH_2_PO_4_	0.5	Sodium lactate (60%)	4
NH_4_Cl	1.0	Yeast extract	1.0
CaCl_2_·4H_2_O	0.1	FeSO_4_·7H_2_O	0.5
Na_2_SO_4_	0.5	Ascorbic acid	0.1
MgSO_4_·7H_2_O	2.0	—	—

**Table 2 microorganisms-13-01511-t002:** Experimental design of CSR.

CSR	Carbon Source Addition (g/L)		
	Sodium Lactate	Glycerin	Ethanol
0.67	2.34 (L1)	0.49 (G1)	0.82 (E1)
1.32	4.67 (L2)	0.98 (G2)	1.64 (E2)
2.01	7.01 (L3)	1.46 (G3)	2.47 (E3)
2.68	9.35 (L4)	1.95 (G4)	3.29 (E4)

**Table 3 microorganisms-13-01511-t003:** Physiological and biochemical characteristics of strain REO-01.

Test	REO-01 Result
Hydrogen sulfide	+
Indole	+
Catalase	+
*N*-acetylglucosamine	+
Oxidase	−
Gram stain	−
Urease	−
Methyl red	−
Sodium pyruvate	+
Glycerol	+
Tartarate	+
Acetate	+
Glucose	−
Mannitol	−
Arabinose	−
Sucrose	−
Malonate	−
Fructose	−
Nitrate reduction	+

Note: “+” indicates a positive result or utilization; “−” indicates a negative result or no utilization.

## Data Availability

The data presented in this study are openly available in NCBI, reference number OL454786.1.

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
