# Peer review of "Optimization of Growth Conditions of Desulfovibrio desulfuricans Strain REO-01 and Evaluation of Its Cd(II) Bioremediation Potential for Detoxification of Rare Earth Tailings"

_microorganisms, 2025, doi:10.3390/microorganisms13071511_

Round 1
Reviewer 1 Report
Comments and Suggestions for Authors
The reviewed article describes the results of an experimental study of the functioning indicators of sulfate-reducing bacteria Desulfovibrio desulfuricans strain REO-01 under different cultivation conditions to determine the optimal conditions. The strain under study was previously isolated by the authors from rare earth tailings and is considered a potential agent for simultaneous sulfate reduction and Cd(â…¡) remediation. Since the isolation process and characterization of the specified strain are given in a previously published article, I consider it inappropriate to indicate in the title of this reviewed article, abstract and research methods that it has been newly identified. It is sufficient to provide a brief description (as in Materials and Methods) with reference. The title of the article should be changed, for example, to the following: Optimization of the growth of the sulfate-reducing bacterium Desulfovibrio desulfuricans strain REO-01 - a promising Cd(II) remediation agent in rare earth tailings. The conducted research is significant in scope and depth, covering the assessment of various indicators - from strain growth to physicochemical indicators of the nutrient medium. While highly appreciating the conducted research and the results obtained, I have some comments:
- The abstract should be rewritten, clearly stating the aim of the study.
- The graphic abstract does not accurately reflect the content of the article. The strain was isolated earlier.
- Line 108 - It is improper to state that the isolation of the strain occurred in a peer-reviewed study.
- What is the name of the culture medium used? Or what culture medium was used as a basis?
- Line 232 - Typically, the pH of the culture medium is adjusted after sterilization using a sterile alkali or acid solution, not before sterilization.
- How were anaerobic conditions created?
- Line 286 - Sulfate-reducing bacteria produce hydrogen sulfide during the reduction of sulfate and other sulfur compounds. How can the obtained data on the positive utilization of hydrogen sulfide be explained? Maybe what was meant here was - positive about the production of this compound? Indole? Catalase? Utilization? Please rewrite the sentence so that there is no ambiguity.
- Lines 418-419 - These are possible chemical transformation reactions. In this study, the products were not determined. You cannot write the name of the strain above the arrow. These reactions are possible transformation pathways. References should be provided.
- The conclusion should be rewritten taking into account the fact that the strain was isolated, identified and described earlier, and this information has already been published and cannot be a generalization of this peer-reviewed article.
- Please cite references according to the journal’s requirements.
- Review the text for technical errors.
After corrections, the article can be published.

Reviewer 2 Report
Comments and Suggestions for Authors
In the manuscript, entitled, "A new single strain of sulfate-reducing bacteria isolated from ion-type rare earth tailings and its characteristics" the results of a comprehensive study of a new strain of sulfate-reducing bacteria for metal bioremediation are published. The authors conducted a detailed study of all the main characteristics of the strain, determined its species, and performed a number of studies on its applied properties for remediation of cadmium and sulfate ions. The text is written competently, the narration is consistent and logical, the conclusions are confirmed by the results. The article presents the results of a large experimental work, which is carried out comprehensively. The graphic design is done at a high level, the number of references to literary sources is sufficient. It is recommended to accept the article after minor changes.
1) Formal design notes. The title is separated from the content of the sub-chapter (line 43). Extra lines, for example, line 270. Incomplete filling of the page with content, for example, pages 11, 13, 15 and others. Figure 1 and its caption are on different pages. On line 418, the illustration is not signed or separated from the text.
2) The authors comprehensively studied the strain of the bacterium, Desulfovibrio desulfuricans. However, despite some references, little attention is paid to the differences between the new strain and the existing ones. For example, the basic cultural properties do not differ, in addition, on lines 334-344, the authors provide data on sulfate reduction, which correspond to the previously described properties of this species, with regard to the time of effective destruction (60 hours). These data are certainly important and confirm the correctness of determining the species of a bacterium, but every time a new species or strain is discovered, the key information is how it differs significantly from those already described earlier.
3) The latin name should be corrected on line 156. The species name desulfuricans should be written with a lowercase letter.
Round 2
Reviewer 1 Report
Comments and Suggestions for Authors
The authors have made corrections to the text of the article in accordance with the comments. The article can be published.